# TEGDMA-Functionalized Dicalcium Phosphate Dihydrate Resin-Based Composites Prevent Secondary Caries in an In Vitro Biofilm Model

**DOI:** 10.3390/jfb13040232

**Published:** 2022-11-09

**Authors:** Andrei Cristian Ionescu, Sebastian Hahnel, Marina D. S. Chiari, Andreas König, Paolo Delvecchio, Roberto Ruggiero Braga, Vanessa Zambelli, Eugenio Brambilla

**Affiliations:** 1Oral Microbiology and Biomaterials Laboratory, Department of Biomedical, Surgical and Dental Sciences, University of Milan, Via Pascal, 36, 20133 Milan, Italy; 2Clinic of Prosthodontics and Dental Materials Science, University of Leipzig, Liebigstraße 12, Haus 1, 04103 Leipzig, Germany; 3Department of Prosthetic Dentistry, Regensburg University Medical Center, Universitätsklinikum Regensburg, Franz-Josef-Strauß-Allee 11, 93042 Regensburg, Germany; 4Department of Biomaterials and Oral Biology, University of Sao Paulo, Av. Prof. Lineu Prestes, 2227, Cidade Universitária, São Paulo 05508-900, SP, Brazil; 5School of Medicine and Surgery, University of Milan-Bicocca, Via Cadore, 48, 20900 Monza, Italy

**Keywords:** bioreactor, secondary caries, caries model, DCPD, micro-CT

## Abstract

This study evaluated the efficacy of experimental TEGDMA-functionalized dicalcium phosphate dihydrate (T-DCPD) filler-based resin-based composites (RBC) in preventing caries lesions around the restoration margins (secondary caries, SC). Standardized Class-II cavities were made in sound molars with the cervical margin in dentin. Cavities were filled with a commercial resin-modified glass-ionomer cement (RMGIC) or experimental RBCs containing a bisGMA-TEGDMA resin blend and one of the following inorganic fractions: 60 wt.% Ba glass (RBC-0); 40 wt.% Ba glass and 20 wt.% T-DCPD (RBC-20); or 20 wt.% Ba glass and 40 wt.% T-DCPD (RBC-40). An open-system bioreactor produced *Streptococcus mutans* biofilm-driven SC. Specimens were scanned using micro-CT to evaluate demineralization depths. Scanning electron microscopy and energy-dispersive X-ray spectroscopy characterized the specimen surfaces, and antimicrobial activity, buffering effect, and ion uptake by the biofilms were also evaluated. ANOVA and Tukey’s tests were applied at *p* < 0.05. RBC-0 and RBC-20 showed SC development in dentin, while RBC-40 and RMGIC significantly reduced the lesion depth at the restoration margin (*p* < 0.0001). Initial enamel demineralization could be observed only around the RBC-0 and RBC-20 restorations. Direct antibiofilm activity can explain SC reduction by RMGIC, whereas a buffering effect on the acidogenicity of biofilm can explain the behavior of RBC-40. Experimental RBC with CaP-releasing functionalized T-DCPD filler could prevent SC with the same efficacy as F-releasing materials.

## 1. Introduction

Resin-based composites (RBCs) are among the most versatile and frequently employed restorative materials in contemporary restorative dentistry [1,2,3]. Though RBC properties have been continuously improved since their introduction, secondary caries is still one of the most frequently observed reasons for the failure of RBC restorations [4,5,6]. Secondary caries is the recurrence of the primary lesion in the natural tissues adjacent to the restoration itself [7]. Indeed, it was estimated that up to 60% of replaced dental restorations belong to a so-called redentistry cycle, which highlights that patients with dental restoration are likely to receive even more dental restorations to repair or replace the original ones [8].

In recent years, biomimetic approaches have been introduced in medicine, including the application of dicalcium phosphate dihydrate (DCPD, brushite, and CaHPO_4_•2H_2_O) in bone cement [9]. Data from the literature show that secondary caries can be considered a material-based problem, as conventional RBCs lack a buffering effect on the acids produced by the overlying biofilm or introduced into the oral environment by the diet [10,11]. Recent innovations in dental materials science in minimizing the prevalence of secondary caries include supplementing restorative materials with DCPD to provide a source of calcium and phosphate ions to the natural tooth tissues adjacent to the restoration when needed [12,13,14]. It has been reported that these ions can reprecipitate to form hydroxyapatite inside carious lesions [15,16], contributing to the remineralization of artificially produced carious lesions [17]. It was also demonstrated that RBCs supplemented with DCPD reduced enamel demineralization [18].

DCPD particles can be functionalized with di- or triethylene glycol dimethacrylate (DEGDMA/TEGDMA) before being incorporated into RBCs to improve the mechanical properties of the material [12]. Data from the literature provide mixed results about their ion release performance. The results of Alania et al. indicated that RBCs featuring functionalized DCPD particles did not reduced the release of ions [13]. On the contrary, Natale et al. found that the ion release from such materials was impaired during the first 15 days [14].

Restorative materials, including RBCs, have to remain in a complex and demanding environment featuring microbial biofilms that permanently colonize oral surfaces [19]. Restorative materials, therefore, can influence biofilm composition and activity. It was shown that conventional RBCs could promote cariogenic biofilm formation by having a surface deprived of buffering capacity [20]. On the contrary, bioactive materials featuring ion release capabilities are considered to mitigate such effects [21]. A previous study from our research group [22] compared biofilm formation on the surfaces of experimental RBCs with triethyleneglycol-dimethacrylate (TEGDMA)-functionalized or non-functionalized DCPD particles (T-DCPD) with conventional RBCs. The results indicated no significant differences in biofilm formation among the tested materials. However, it was shown that DPCD-containing RBCs displayed a relevant buffering effect. Such buffering effects may modulate the interactions between biofilm, natural tooth tissues, and restorative material, reducing cariogenicity and possibly reducing the occurrence of secondary caries.

Thus, the current study aimed to evaluate the effect of two experimental resin-based composites supplemented with a different amount of T-DCPD particles on secondary caries formation using a bioreactor-based microbiological model that produced a cariogenic challenge. The null hypothesis was that the experimental DCPD-containing RBCs would not influence secondary caries formation.

## 2. Materials and Methods

In the first part of the study, standardized disks were made from experimental and control restorative materials. Materials were characterized using SEM, XRD, and thermogravimetric (TGA) analyses. Microbiological behavior, ion uptake by the biofilms, and response to an acidic challenge were also evaluated. In the second part of the study, the experimental and control materials were used to restore standardized class II cavities made on human molars. These specimens were exposed to a secondary caries model using a bioreactor to grow cariogenic biofilms. Secondary caries formation was assessed using micro-CT (Figure 1).

### 2.1. Test Materials

The present study tested three experimental RBCs and a conventional resin-modified glass ionomer cement (RMGIC, positive control; Ionolux, VOCO GmbH, Cuxhaven, Germany). The experimental materials had the same resin matrix composition, consisting of equal molar parts of glycidyl dimethacrylate bisphenol-A (BisGMA) and triethylene glycol dimethacrylate (TEGDMA), including camphorquinone (0.5 wt.%) and N, N-dimethyl aminoethyl methacrylate (0.5 wt.%, all chemicals used in composites formulations were from Merck KGaA, Darmstadt, Germany). They differed in filler composition but not in their filler-to-resin ratio, as follows:-60 wt.% barium glass and 0 wt.% T-DCPD (RBC-0),-40 wt.% barium glass and 20 wt.% T-DCPD (RBC-20),-20 wt.% barium glass and 40 wt.% T-DCPD (RBC-40).

T-DCPD particles were obtained according to the procedure described by Rodrigues et al. [12]. Briefly, DCPD particles were produced via the stoichiometric reaction between aqueous solutions of 78 mmol/L ammonium phosphate, (NH_4_)_2_HPO_4_, and 78 mmol/L calcium nitrate, Ca(NO_3_)_2_•4H_2_O, using the sol-gel approach. Using a peristaltic pump (9 mL/min), 400 mL of the calcium nitrate solution was added drop-wise to the same amount of ammonium phosphate solution, to which 7 g of TEGDMA had previously been added. At room temperature, precipitation occurred under steady stirring, being maintained for 30 min after the dripping ended. The DCPD-TEGDMA interface formed on the particles is represented in Figure 2. After decanting, T-DCPD particles were washed in water to remove ions and excess TEGDMA. The resultant paste was freeze-dried to produce a white powder.

### 2.2. Preparation of Disk Specimens for Surface Characterization and Biofilm Formation

A total of 24 disks were prepared from each tested restorative material, as described previously [22]. For the preparation of a single specimen, a standardized amount of uncured RBC/RMGIC was packed into a custom-made PTFE mold with a diameter of 6.0 mm and a height of 2.0 mm, covered with a cellulose-acetate strip, condensed against a microscopic glass slide, and light-cured using an LED curing unit (Radii cal, SDI, Bayswater, Australia; 1200 mW/cm^2^ curing power for 40 s). Round enamel disks with identical dimensions were cut from the vestibular surfaces of permanent human teeth extracted for clinical reasons with a water-cooled trephine bur (INDIAM, Carrara, MS, Italy). All disks were polished using diamond burs and silicon carbide sheets (600, 1200, 2000 grit) before additional processing to achieve peak-to-valley surface roughness (Ra) values of approximately 0.2 μm, measured with a profilometer (Surtronic 3+, Taylor-Hobson, Leicester, UK). All specimens were sterilized using a chemiclave (Sterrad, ASP, Irvine, CA, USA), placed in artificial saliva for one week, and labeled “1W”. The artificial saliva was changed daily to eliminate leachates (unpolymerized monomers) and avoid a burst effect of ions in the initial few days. The artificial saliva was made to mimic the specific electrolyte content of whole human saliva and was prepared from 0.1 L of 150 mM KHCO_3_, 0.1 L of 100 mM NaCl, 0.1 L of 25 mM K_2_HPO_4_, 0.1 L of 24 mM Na_2_HPO_4_, 0.1 L of 15 mM CaCl_2_, 0.1 L of 1.5 mM MgCl_2_, and 0.006 L of 25 mM citric acid. The volume was made up to 1 L, and the pH was adjusted to 7.0 by pipetting 4 M NaOH or 4 MHCl solutions with vigorous stirring [23].

### 2.3. Preparation of Tooth Specimens for Secondary Caries Evaluation

This study was conducted according to the guidelines of the Declaration of Helsinki and approved by the Ethics Committee of the University of Milan (protocol code SALTiBO-2017).

As the degree of enamel and dentine mineralization is one of the factors associated with the variability of results when using natural substrates, data from a preliminarily performed pilot study were used to determine the sample size [24].

According to demineralization depth data gathered in that study, the sample size was determined considering the teeth as the primary source of variability, with a standard deviation of the demineralization depth of 25 µm, an α value of 0.05, and a power of 0.8. A total of 7 teeth were estimated to be necessary. Therefore, a total of eight teeth were used in the present study.

The Oral Surgery Unit of the Department of Biomedical, Surgical, and Dental Sciences (Milan, Italy) provided eight sound human molars extracted for clinical reasons. Their roots were severed at a 3 mm apical distance from the cementoenamel junction (Isomet 1000, Buehler, Lake Bluff, IL, USA). The pulp was removed, and pulp chambers were filled with a regular RBC restoration (Majesty ES-2 RBC bonded with Clearfil SE Bond, Kuraray Noritake, Tokyo, Japan). A total of four standardized class II cavities (4 mm width, 2 mm depth, and 6 mm height), one for each side, were prepared on each tooth specimen using 2 mm-wide cylindrical diamond burs. The cervical margins of the cavities were in the dentin (Figure 3A). The cavities of each tooth were randomly filled using the tested materials, then light-cured as specified. To focus on the effects of the composite without interference from the adhesive, no adhesive was used [15,25,26]. Before further processing, all specimens were finished using diamond burs and silicon carbide papers to produce Ra values ≈ of 0.2 µm. (Surtronic 3+) to ensure standardization of this parameter. All specimens were placed in artificial saliva for one week and then sterilized (Sterrad).

### 2.4. Characterization of Test Materials by Scanning Electron Microscopy (SEM) and Energy-Dispersive X-ray Spectroscopy (EDS)

A total of four disks for each material underwent SEM/EDS analyses using a tabletop scanning electron microscope (TM4000Plus, Hitachi, Schaumburg, IL, USA) equipped with an EDS probe (Quantax 75 with XFlash 630H Detector, Bruker Nano GmbH, Berlin, Germany). Dry specimens were mounted on stubs using conductive tape and analyzed without sputter-coating, using the secondary electron (SE) and backscattered electron (BE) detector at an accelerating voltage of 15 kV in a surface-charge reduction mode. The surface morphology of the tested materials were thus analyzed, with greater attention on filler particle distribution, shape, and dimensions. In particular, the BE mode that was sensitive to an element’s atomic number enabled differentiation between fillers of different compositions, making elements with higher atomic numbers appear whiter. Three randomly selected micrographs were obtained for each disk (500×, 2000×, and 5000×) for morphological observation of the surface and with the EDS probe (300 × 300 μm fields) in full-frame mode at 150 s acquisition time. The data acquired by EDS were averaged for each specimen and element, representing the wt.% distribution in a ≈1 μm superficial layer (See Ionescu et al., 2022, Figure 6 [27]). Elemental distribution at the surface level was visually displayed using the map mode (5000×), with an acquisition time of 600 s.

### 2.5. X-ray Diffraction

Non-destructive X-ray diffraction (XRD) analysis was used to detect and identify crystalline phases in the filler on the surface of the disks. A total of three specimens were analyzed. The measurements were performed using an X-ray diffraction platform (D8 DISCOVER, Bruker, CuKα, 40 kV, 40 mA) equipped with a 2D VÅNTEC-500 (Bruker) detector. The averaged observed intensity was plotted against the position (2Ɵ), and diffraction peaks were searched in the resulting graph.

### 2.6. Thermal Analysis (Thermogravimetry, TGA)

A simultaneous apparatus for thermogravimetric analysis and differential scanning calorimetry (TG-DSC, STA 449, Netzsch GmbH & Co, Selb, Germany) was applied to quantify the free water content and amount of DCPD in the experimental disks (three specimens per each material). A 5 K/min heating rate under an air atmosphere and a maximum temperature of 1.000 °C were used. Temperatures were recorded and averaged for each material.

### 2.7. Microbiological Procedures

#### 2.7.1. Saliva Collection

The experimenters provided paraffin-stimulated whole saliva obtained according to a previously reported methodology [28]. Participants did not brush their teeth for 24 h, had no current dental disease, and did not take antibiotics for at least three months. Saliva was collected in refrigerated test tubes before the onset of the COVID-19 epidemic, heated to 60 °C for 30 min, then centrifuged (12,000× *g*, 4 °C, 15 min). The supernatant was then transferred to sterile tubes and kept at −80 °C before being thawed at 37 °C for 1 h before use.

#### 2.7.2. Bacteria

Cultures of *Streptococcus mutans* (ATCC 35668) were prepared, as previously described [23]. Agar plates (Mitis Salivarius Bacitracin) were cultured at 37 °C in a 5% CO_2_ enriched atmosphere for 48 h. After overnight incubation at 37 °C in the 5% CO_2_ atmosphere, a pure suspension of the microorganisms was obtained in brain heart infusion (BHI), supplemented with 1 wt.% sucrose. Centrifugation (2200× *g*, 19 °C, 5 min) was used to collect the cells, which were then washed twice with phosphate-buffered saline (PBS) and resuspended in the same buffer. After that, the suspension was treated with low-intensity ultrasonic energy (Sonifier model B-150; Branson, Danbury, CT, USA) to disperse bacterial chains before being adjusted to a McFarland scale equivalent of 1.0, corresponding to a microbial concentration of roughly 6.0 × 10^8^ cells/mL.

#### 2.7.3. Bioreactor

The disks were placed into the flow cells of a continuous-flow bioreactor system (modified drip-flow reactor, MDFR, Figure 3B) [29]. The modified design included placing the bioreactor in a horizontal position. It allowed the placement of customized polytetrafluoroethylene (PTFE) trays containing 27 holes, in which each specimen was tightly fixed on the bottom of the flow cell, exposing its surface to the surrounding medium. The MDFR’s tubing and specimen-holding trays were sterilized before tests (Sterrad). The entire setup was then placed within a sterile hood. Each flow cell was filled with 10 mL of thawed sterile saliva to promote the development of a salivary pellicle on the surface of the tested disks (n = 12 for each material). The MDFR was incubated at 37 °C for 24 h; then, the supernatant saliva was discarded. *S. mutans* biofilm formation was initiated by filling the flow cells with a 1:1 dilution of the bacterial culture in 1 wt.% sucrose-supplemented BHI. After a 4 h incubation under static conditions at 37 °C in the 5% CO_2_ atmosphere, a multichannel, computer-controlled peristaltic pump (RP-1, Rainin, Emeryville, CA, USA) started delivering a continuous flow (20 mL/h) of sterile modified artificial saliva [28]. The latter was made of 10.0 g/L sucrose, 2.5 g/L mucin (type II, porcine gastric), 2.0 g/L bacteriological peptone, 2.0 g/L tryptone, 1.0 g/L yeast extract, 0.35 g/L NaCl, 0.2 g/L KCl, 0.2 g/L CaCl_2_, 0.1 g/L cysteine HCl, 0.001 g/L hemin, and 0.0002 g/L vitamin K_1_. The MDFR was operated for 48 h to develop a multilayer biofilm on the specimen surfaces. At the end of the incubation, the flow was stopped, and trays were extracted from the flow cells. The disks were carefully removed from the trays using a pair of sterile tweezers and then gently rinsed with sterile phosphate-buffered saline (PBS) at 37 °C to remove non-adherent cells. The disks were then placed into sterile 48-well plates. The adherent, viable biomass was assessed on half of each material’s disks. The other half were sonicated (Sonifier Model B-15 operating at 40 W energy output for 10 min) and carefully cleaned using a microbrush to remove any biofilm. Then, they were labeled as “BIO” and subjected to the XRD and TG-DSC analysis, as previously described, to evaluate the effect of biofilm formation on the tested materials.

Additional disks (labeled as “ACID”) were incubated in the bioreactor using the same setup as before, keeping the disks under sterile conditions for the entire incubation time. The filter-sterile effluent broth collected from the previous bioreactor runs with BIO specimens was used to flow the cells, thus exposing the disks to the same pH values generated by the *S. mutans* biofilms (pH = 4.2 ± 0.1).

#### 2.7.4. Viable Biomass Assessment

The viable biomass assessment was performed as previously described [23]. Briefly, two stock solutions were prepared by dissolving 5 mg/mL of 3-(4,5)-dimethylthiazol-2-yl- 2,5-diphenyltetrazolium bromide (MTT) and 0.3 mg/mL of N-methylphenazonium methyl sulphate (PMS) in sterile PBS. The solutions were stored at 2 °C in light-proof vials until the day of the experiment, when a test solution (TS) was prepared by mixing MTT stock solution, PMS stock solution, and sterile PBS in a 1:1:8 ratio. A lysing solution (LS) was prepared by dissolving 10 vol% sodium dodecyl sulfate and 50 vol% dimethylformamide in distilled water. TS and LS were brought to 37 °C before use. A total of 300 μL of TS was placed in each well of the 48-well plates containing the test disks. The plates were incubated at 37 °C under light-proof conditions. During incubation, electron transport across the microbial plasma membrane and, to a lesser extent, microbial redox systems converted the yellow salt to insoluble purple formazan crystals. The intermediate electron acceptor (PMS) facilitated the conversion at the cell membrane level. After one hour, the TS solution was carefully removed, and 300 μL of LS was added to each well. The plates were then stored under light-proof conditions for one additional hour (room temperature) to allow the dispersion of the formazan crystals into the surrounding solution. Subsequently, 100 μL of the supernatant was transferred to a 96-well plate, and the absorbance was measured at a wavelength of 550 nm using a spectrophotometer (Genesys 10-S, Thermo Spectronic, Rochester, NY, USA). Results were expressed as relative absorbance in optical density (OD) units corresponding to adherent, viable, and metabolically active biomass.

### 2.8. Supernatant pH Changes Induced by Acidic Challenge

A total of four randomly selected disks for each material were incubated with 500 μL of filter-sterile effluent broth (pH = 4.2) in 48-well plates at 37 °C and 100 rpm in a rotary shaking incubator for 48 h, after which the pH of the solutions was measured at room temperature using a ROSS pH electrode (Thermo Scientific, Waltham, MA, USA).

### 2.9. Determination of Ion Content in Biofilms Grown on Test Disks

The detached biofilm from BIO specimens was collected, and the ions eluted from the experimental disk surfaces and diffused into the biofilms were determined. After centrifugation (2200× *g*, 19 °C, 5 min), biofilms were diluted in 1 mL of a 10 vol% HNO_3_ solution for 10 min. After thorough stirring, the volume was made up to 10 mL by adding distilled water, vials were centrifuged (2200× *g*, 19 °C, 5 min), the supernatant was recovered, and the pellet was weighed to determine wet biomass weight. Al, Sr, and Ca concentrations were determined by inductively coupled plasma mass spectrometry (ICP-MS, NexION 350X, PerkinElmer, Waltham, MA, USA) in kinetic energy discrimination mode. The instrument was operated using helium as the collision cell gas. Quantitative analysis was performed using calibration with Re^187^ and Ge^72^. A total of 5 mL of the test solutions were diluted with 5 mL 1-butanol, 2 mL Triton-X, and 5 g of EDTA, and the volume was made up to 1 L with ultrapure water. Ion-selective electrode (ISE) analysis was used to determine F^−^ concentrations (F-ISE), as described [23]. In brief, a stock solution with a fluoride concentration of 1000 ppm was diluted in 1 vol% HNO_3_ to obtain fluoride standards with fluoride concentrations ranging from 0.0019 to 64 parts per million (ppm). A calibration curve was obtained by measuring fluoride standards using a digital pH/mV meter (SA-720, Orion Research Inc., Boston, MA, USA). Before the analyses, the sodium acetate buffer 20% *v*/*v* with EDTA as an ionic strength adjustor was added to each standard. A negative reference standard (0 ppm fluoride) was prepared by adding 20% *v*/*v* of sodium acetate buffer with EDTA to the 1 vol% HNO_3_ solution; this solution was also used to rinse the electrodes between measurements.

### 2.10. Secondary Caries Models

The eight restored tooth specimens were placed into the flow cells of the previously described continuous flow MDFR bioreactor system (Figure 3B, [24]). *S. mutans* biofilm formation was initiated by filling the flow cell with the bacterial suspension diluted 1:1 with sucrose-supplemented BHI. After overnight incubation under static conditions at 37 °C in the 5% CO_2_ environment, the peristaltic pump flowed the sterile modified artificial saliva for two weeks.

The pH values of the effluent culture broth, as well as the absence of any contamination, were controlled daily. The pH of the broth was measured at room temperature using a ROSS micro-pH electrode, while three drops of effluent broth were pipetted on a glass slide, GRAM-stained, and observed under a microscope.

### 2.11. Microtomography

Restored tooth specimens were scanned with a microCT (Skyscan 1176, Bruker, Kontich, Belgium; 9 µm resolution, 80 kV, 300 mA) before (Figure 4) and after biofilm formation. After two weeks of incubation, the restored specimens were carefully extracted from the two bioreactor systems and placed in a Falcon vial under 100% relative humidity until the microCT scans were completed. The acquisition parameters were as follows: 80 kV, 300 µA source current, 8.92 µm isotropic pixel size, 0.5 mm Al + 0.38 mm Cu filter, and a 0.5° rotation step over 360° with a frame averaging of 5. Reconstruction was performed using proprietary software (NRecon, Bruker) with standardized parameters of beam hardening correction (30%), smoothing = 7, smoothing kernel = 2 (Gaussian), ring artifact compensation = 20, and optimal contrast limits based on the initial scanning and reconstruction tests. Image data were then aligned and resliced using another proprietary software (Dataviewer v1.5.4.6, Bruker). Demineralization depths (µm) were evaluated directly at the margins and 1.0 mm from the restoration interface at 16 randomly-selected sites of each restoration (8 sites at the enamel-restoration interface and 8 sites at the dentin-restoration interface, with 8 on the right side and 8 on the left side of each restoration) [24]. A single operator blinded to the experimental groups defined the thickness of the demineralized layer that appeared more radiolucent than the underlying sound tissues for each site. The presence of CT artifacts due to material radiopacity and air bubbles trapped inside materials prevented us from applying an automatic AI-based threshold detection method.

### 2.12. Statistical Analysis

Statistical software (JMP 16.0, SAS Institute, Cary, NC, USA) was used to perform all statistical analyses. Normal data distribution was checked using Shapiro–Wilk’s test, and the homogeneity of variances was verified using Levene’s test. Means and standard errors were calculated from the raw data. Multi-way analysis of variance (ANOVA) was used to analyze datasets. Gap data were analyzed considering material and distance from the interface as fixed factors. MTT/PMS, buffering capacity, and ion content datasets were analyzed using one-way ANOVA and Tukey’s HSD post hoc test to highlight significant differences between materials. The level of significance (α) was set to 0.05.

## 3. Results

### 3.1. Materials Surface Characterization (SEM-EDS, XRD, and TGA)

T-DCPD particles were displayed by SEM, having the shape of irregular microplates and a mean size of 10 μm (Figure 5). Such microplates had a dark grey signal indicating a composition of lightweight elements (Ca and P), as opposed to the Ba glass microparticles of the inert filler (mean size 1 μm). They were more abundant in RBC-40 than in RBC-20 and absent in RBC-0, in accordance with the material preparation. RMGIC showed filler particles with the same shape and size as the Ba glass but with a more widespread particle size distribution. The latter material also showed a higher organic-to-inorganic ratio. EDS analysis of the surface (Figure 6) confirmed experimental RBCs made of a Si-Al-Ba glass filler, to which Ca-P microplates were added in RBC-20 and RBC-40. The Ca-P surface content on RBC-40 was three times that of RBC-20. RMGIC was made of F-Al-Si glass with the addition of Sr and traces of Na, P. EDS analysis of ACID and BIO disks (Figure 7) showed a considerable decrease in surface Ca and P content, whereas no changes were observed in RBC-0 disks.

Furthermore, BIO disks displayed an increase in surface carbon content. RMGIC disks also underwent substantial changes in surface elemental composition, with RMGIC-ACID specimens showing depletion of F and Sr and a relative increase in carbon content. Moreover, RMGIC-BIO specimens displayed a decrease in carbon content compared with untreated disks (1W) and only a slight decrease in F and Sr.

XRD identified an amorphous pattern with no crystalline phases in all RBC-0 disks (Figure 8). Immediately after storage (1W), peaks corresponding to crystalline DCPD were identified on the surface of RBC-20 and RBC-40, having a higher surface content of DCPD in RBC-40 than in RBC-20. Interestingly, the anhydrous Ca-P form of DCPD called monetite was also identified on RBC-20 and RBC-40 disks subjected to biofilm formation or acidic conditions (Table 1). A higher amount of monetite was detected on RBC-40 surfaces than on RBC-20, consistent with Ca-P presence.

TGA was used to measure the water content of the tested materials, defined as the loss of mass between 20 and 105 °C (Table 2 and Figure 9). Very low overall amounts of free water content were detected, with the maximum free water content found in RBC-20 after storage (1W, 0.5 wt.%).

The content of DCPD in the various specimens was stoichiometrically calculated based on the three-step dehydration process between 105 and 230 °C (Rabatin et al., 1960; Dosen and Giese, 2011), based on the following reaction: Ca[PO_3_(OH)]·2H_2_O → Ca[PO_3_(OH)] + H_2_O
172.09 → 136.06 + 36.03 (g/mol)

Based on these calculations, the DCPD content of the specimens was estimated to be 22.1 wt.% in RBC-20 and 40.2 wt.% in RBC-40. Biofilm formation greatly reduced DCPD content in both T-DCPD-containing materials, whereas a much smaller decrease was observed in specimens subjected to acidic conditions (ACID).

### 3.2. Antibacterial Activity

The lowest amount of viable biomass was found on RMGIC surfaces (*p* < 0.0001), indicating that this material exhibited significant antimicrobial activity. Similar amounts of viable biomass were identified on RBC-40 and enamel. RBC-0 and RBC-20 showed intermediate values between enamel and RMGIC (Figure 10).

### 3.3. Supernatant pH Changes Induced by Acidic Challenge

The tested materials tended to increase pH values: enamel > RMGIC > RBC-40 > RBC-20. No significant change in pH values was demonstrated for RBC-0 (*p* = 0.5568, Table 3).

### 3.4. Determination of Ion Content

After weighing the wet biomass, data of ion content were converted into ng/mg of wet weight and log-transformed to achieve normal distribution. ICP-MS and F-ISE analysis (Figure 11) showed the presence of Ca^2+^ inside the biomass developed on all specimen surfaces. However, the calcium content was significantly higher in biofilms growing on enamel surfaces (*p* < 0.0001). A significantly higher (two orders of magnitude) presence of F^−^ was identified in biofilms growing on RMGIC than in the other materials, whereas biofilms growing on enamel and RBC-40 showed a slight but significantly higher fluoride content than RBC-20 and RBC-0. Biofilms growing on RMGIC showed the presence of Al and Sr ions.

### 3.5. Secondary Caries Models

Micro-CT analysis before the cariogenic challenge (T = 0) showed no evidence of microgaps in any restoration, despite using no adhesive system. The average pH of the culture broth measured in the bioreactor was 4.2 ± 0.1, and no contamination was found during the experiments. Micro-CT identified dentin demineralization in all specimens (Table 4, Figure 12). Cavities restored with RBC-40 and RMGIC showed significantly lower demineralization depths at the restoration margins compared with cavities restored with RBC-20 and RBC-0. No significant difference was found in demineralization depths 1 mm from the restoration margins. Initial lesions on enamel were identified in 50% of the sites of RBC-20 and RBC-0. These lesions were found to have a mean depth of 50 µm. Micro-CT did not identify secondary caries lesions on enamel close to RBC-40 and RMGIC margins (Figure 13).

## 4. Discussion

Researchers have been trying to overcome the issue of secondary caries formation for a long time. It is known that dental materials surfaces that do not provide a buffering effect can lead to the selection of cariogenic flora on the overlying biofilm [20]. On the other hand, bioactive materials such as glass-ionomer cements and their resin-modified counterparts often do not adequately meet the mechanical and aesthetic properties required to replace compromised hard tissues [30]. Filling an RBC material with a source of calcium and phosphate ions may be a good choice in this sense, providing a material that may reduce secondary caries. This material may still need improvements in its mechanical, physical, and aesthetical properties [14,16]. It is challenging to manipulate the refractive indices of crystalline calcium phosphates, which hinders achieving translucency and pleasing final aesthetics [31]. In this study, we evaluated the effect of two experimental RBCs supplemented with DCPD particles on the formation of secondary caries in a bioreactor model simulating an acidic challenge produced by cariogenic biofilms. Based on our findings, we could reject our null hypothesis stating that the experimental DCPD-containing RBCs would not influence the formation of secondary caries. The use of an experimental RBC filled with 40 wt.% of TEGDMA-functionalized DCPD significantly reduced demineralization depths in the dentinal tissues in immediate proximity to the restoration, whereas no demineralization was observed close to the enamel. A threshold value seems to exist as the 20 wt.% T-DCPD-filled RBC did not show such activity.

Interestingly, despite having a very different formulation and activity, RBC-40 obtained similar protection against secondary caries formation as the tested RMGIC. Fluoride-releasing restorative materials, such as the reference RMGIC, are widely used in dentistry due to their anticaries effect [32,33]. Fluoride ions released from the material can directly inhibit the metabolic pathways of microbial cells in the overlying biofilm and accumulate on surrounding hard tissue surfaces [34]. The latter leads to the precipitation of calcium fluoride-like deposits, which act as a reservoir and gradually release fluoride during decreased pH conditions [27,35]. The released fluoride prevents enamel demineralization and reduces caries susceptibility. Incorporating fluoride into enamel hydroxyapatite via filling or displacing the hydroxyl vacancies stabilizes the crystal structure that forms fluorapatite and lowers the solubility product constant (Ksp). Consequently, fluoride-treated dental tissues show increased demineralization resistance and decreased biofilm acidogenicity. The results of the present study confirm these mechanisms of action as biofilms growing over RMGIC surfaces contained fluoride ions and showed significantly lower viable biomass when compared with the ones growing on a conventional RBC. The material was also able to buffer an acidic pH, and the ion depletion observed by the surface analysis showed that this activity was due to the ion release. The downside of this activity was the relatively swift ion depletion at the surface level, as shown by SEM-EDS. The acidic challenge under sterile and biofilm formation conditions caused a steep decrease in F, Al, Si, and Sr concentrations at the material’s surface, limiting its ion release capability. Though it has been demonstrated in vitro that fluoride recharge can extend the ion release capability of such material over time [23], the extent to which such a recharge can prolong the caries prevention of RMGIC remains to be investigated.

Brushite can be transformed into monetite by hydrolysis, sol-gel, and hydrothermal methods when DCPA cements are set at a very low pH, in water-deficient environments, or the presence of metallic ions disrupt brushite crystals, favoring monetite formation [36]. When RBC-40 was challenged with biofilm formation, XRD data showed that part of the brushite was converted into monetite. The same effect was seen when RBC-40 was challenged with a sterile run with bacteria-produced acids, suggesting that the conversion was made possible at 37 °C by the acidic environment caused by the cariogenic biofilm. On the other hand, the thermogravimetric analysis showed that a significant reduction in DCPD content occurred during the biofilm challenge. This reduction was higher on biofilm-challenged specimens than on acid-challenged specimens. These results suggest that DCPD was partly eluted from the experimental RBC surface and was partly converted into DCPA (dicalcium phosphate anhydrous, monetite) because of bacterial activity that was not limited to acid production. 

Several processes in which DCPA cements are exposed to excessive or even moderately acidic conditions result in the precipitation of monetite instead of brushite during the cement setting reaction. Monetite is more stable and undergoes slower degradation than brushite; its nucleation and crystal growth are much less energetically favored than brushite [37]. Therefore, brushite may have decomposed to monetite, or the released calcium and phosphate ions may have reprecipitated under acidic conditions into monetite. Significant amounts of the surface calcium phosphate remaining on T-DCPD-containing RBCs after acidic and biofilm formation challenges were monetite, implying lower bioavailability of such ions due to the higher stability of monetite. However, residual brushite and monetite crystals found by XRD may help in the epitaxial growth of micrometric Ca-hydroxyapatite crystals over time on RBC surfaces [34], helping achieve a bioactive and biomimetic surface coating on the tested experimental RBCs [28]. Further studies are needed to investigate such possibilities.

The tested T-DCPD-containing RBC-40 may have had three possible mechanisms by which it showed an influence on secondary caries reduction: (i) Direct antimicrobial effect of the material, causing lower biomass and catabolite production, leading to fewer secondary caries; (ii) Ion release providing a buffering effect of the acidic catabolites of the biofilm, reducing their demineralizing activity; and (iii) Direct effect of the released ions in remineralizing initial secondary caries lesions. Therefore, additional experiments were performed to ascertain the contribution of each of these mechanisms to the reduction of secondary caries formation.

Antimicrobial test results showed no significant antibacterial effect of the tested DCPD-containing RBCs compared with enamel surfaces, whereas a significant effect was expressed by the fluoride- and strontium-releasing RMGIC material. Melo et al. used a human in situ model to test biofilm formation on experimental RBCs containing nano amorphous calcium phosphate (nACP). They did not identify any significant differences in biofilm formation when comparing this experimental composite with a conventional RBC [38]. It should be noted that the surface area of nACP is five to eleven times higher than that of other calcium orthophosphate particles, such as DCPD.

For this reason, Rodrigues et al. (2015) found that nACP-filled experimental RBCs provided a higher ion release [39]. If this were related to antimicrobial activity, a high activity would be expected from nACP-filled materials. Furthermore, a previous study from our research group did not identify any significant anti-biofilm activity of both DCPD or functionalized T-DCPD-filled experimental RBCs [22], even though a previous study found that TEGDMA functionalization significantly decreased ion release from DCPD-filled experimental RBCs [13]. Therefore, the surfaces of the calcium orthophosphate-containing experimental RBCs or the ions released from the surfaces of such materials were not shown to reduce biofilm formation directly.

A buffering capacity by ion release is a very interesting possibility for bioactive/biomimetic restorative materials to control biofilm cariogenicity. A recent study showed that materials such as conventional RBCs that lack buffering capacity could promote shifts in oral biofilm composition toward cariogenic species [20]. Such species, which have an acid-tolerant phenotype and a high acid production rate, are mainly responsible for developing secondary caries. Our results showed that both the tested T-DCPD-containing RBCs and reference RMGIC had significant buffering capacities. The capacity was highest for RBC-40 and RMGIC but was still significantly lower than the buffering capacity of enamel. The buffering capacity expressed by DCPD-containing RBCs due to ion release has been confirmed by several studies [15,22,39] and is consistent with the results obtained in the present study using the secondary caries microbiological challenge.

The results of the present study must be considered with its limitations, including the use of experimental RBCs without conditioning of the tissues or application of an adhesive procedure. Micro-CT analysis before the cariogenic challenge (T = 0) excluded the presence of micro-gaps that may have been caused by not using such procedures. This choice was made to focus on the effects of the experimental composite without interference from an adhesive. Furthermore, the use of the experimental RBCs without an adhesive procedure followed similar application procedures [15,25,26]. Further experiments should make use of an adhesive procedure for a closer approximation of clinical conditions.

## 5. Conclusions

The results of the current study indicate that RBCs with a fraction of 40 wt.% TEGDMA-functionalized DCPD filler particles show promising bioactive properties that reduce the development of secondary caries in vitro by buffering the acidogenicity of the biofilm. The anticaries activity was similar to a fluoride-containing conventional resin-modified glass-ionomer cement restorative material.

## Figures and Tables

**Figure 1 jfb-13-00232-f001:**
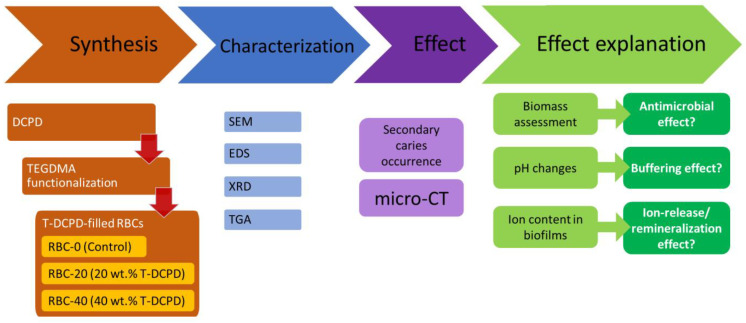
Flowchart illustrating the experimental setup. Triethyleneglycol-dimethacrylate (TEGDMA)-functionalized dicalcium phosphate dihydrate (DCPD)-filled resin-based composites (RBCs) were obtained. Then, materials were extensively characterized, and their activity was evaluated in terms of their ability to prevent/reduce secondary caries occurrence. Additional experiments were performed to explain such an activity, as shown. SEM, scanning electron microscopy, EDS, energy-dispersive x-ray spectroscopy, XRD, x-ray diffraction, TGA, thermogravimetry, micro-CT, micro-computed tomography.

**Figure 2 jfb-13-00232-f002:**
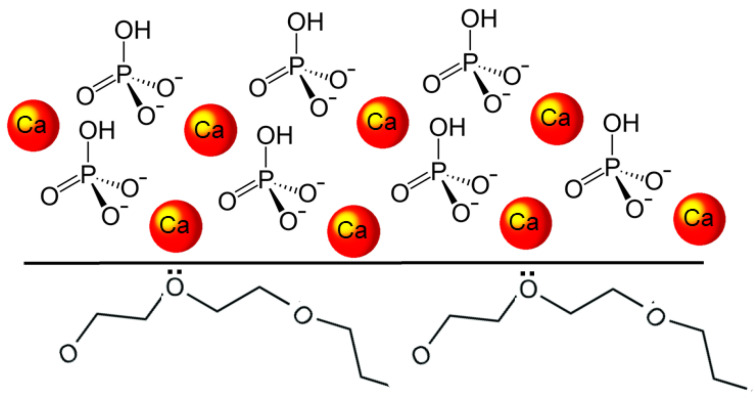
Diagram illustrating the DCPD-TEGDMA interaction. The DCPD structure is represented by the calcium ions and hydrogen phosphate tetrahedra. At the DCPD-TEGDMA interface, two triethylene glycol groups from TEGDMA are represented, with ion-dipole bonds established between one of their oxygen atoms and the calcium ion.

**Figure 3 jfb-13-00232-f003:**
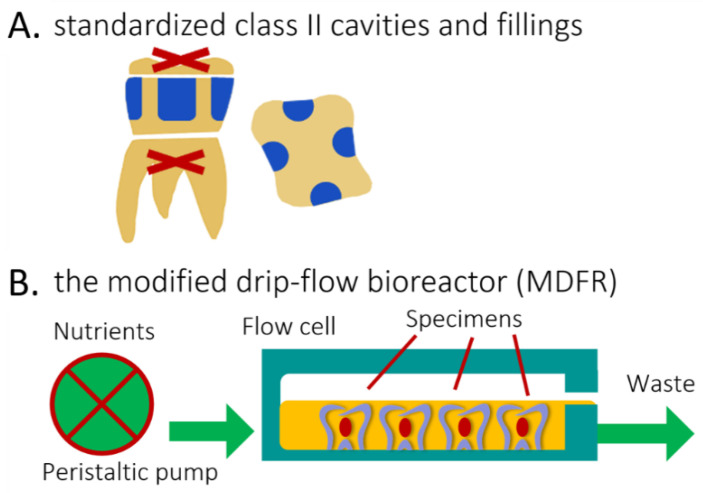
Explanation of the methods and procedures. (**A**), Lateral and top view of de-rooted human molars each harboring four standardized class II cavities (in blue, 4 mm width, 2 mm depth, and 6 mm height), with cervical margin in dentin. (**B**) Diagram of the open system bioreactor based on a drip-flow bioreactor. Specimens are depicted covered by the nutrient broth (orange).

**Figure 4 jfb-13-00232-f004:**
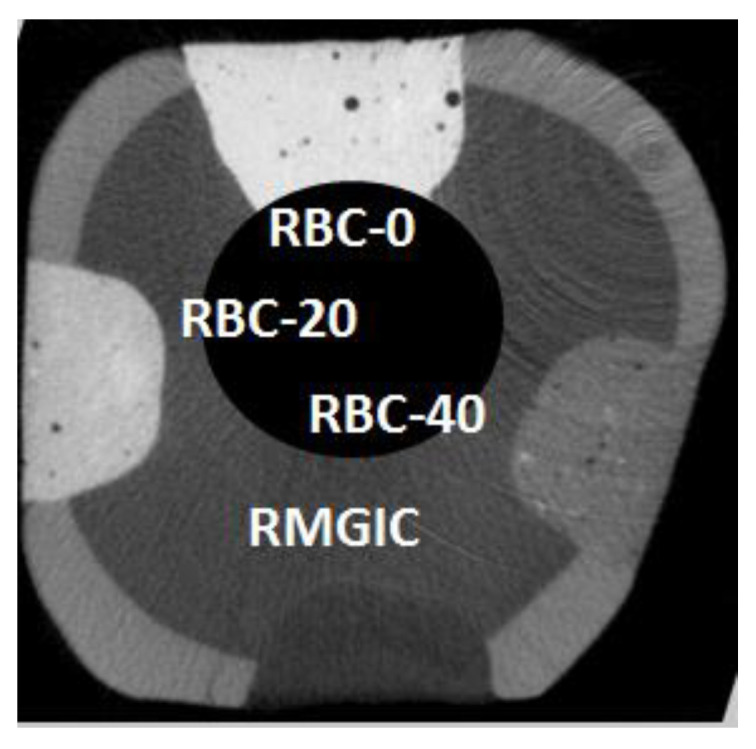
Example of a coronal section from a micro-computed tomography CT scan before incubation (t = 0), displaying a tooth specimen with the four restorations from the tested restorative materials at enamel level. The barium glass filler was responsible for increased radiopacity that was highest in RBC-0, corresponding to 60 wt.% Ba glass, and lower in RBC-40, having only 20 wt.% Ba glass. RMGIC had even lower radiopacity. In this way, materials could be immediately identified on micro-CT scans. Cavity dimensions are 4 mm wide and 2 mm deep.

**Figure 5 jfb-13-00232-f005:**
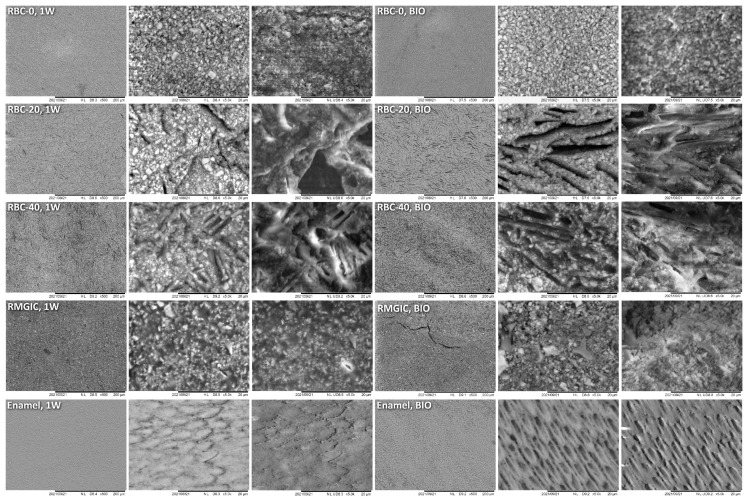
SEM characterization of the tested materials. Surfaces of specimens after 1-week storage in artificial saliva (1W) were compared with the ones produced by 48-h incubation with an *S. mutans* biofilm under constant acidic challenge.

**Figure 6 jfb-13-00232-f006:**
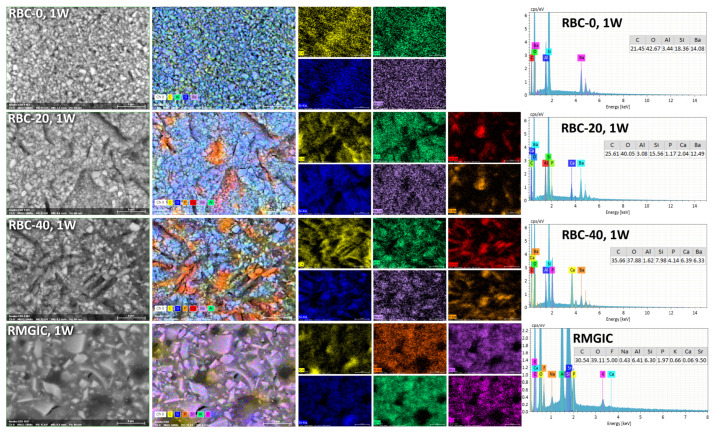
SEM-EDS analysis of the tested materials after procedures and storage for one week in artificial saliva.

**Figure 7 jfb-13-00232-f007:**
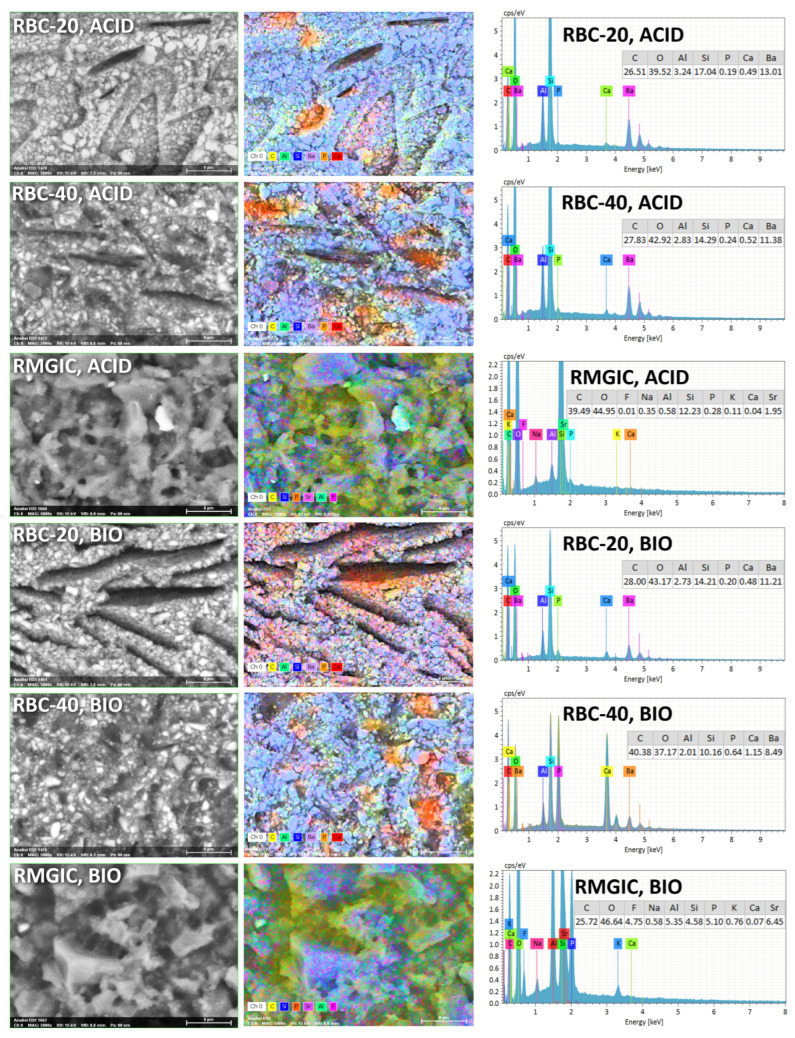
SEM-EDS analysis of treated materials (ACID and BIO treatments, indicating exposure in the bioreactor to acidic conditions or an acidogenic biofilm, respectively, for 48 h).

**Figure 8 jfb-13-00232-f008:**
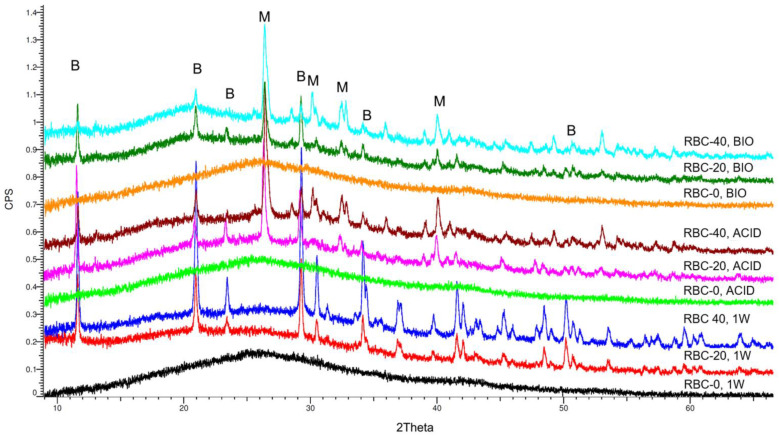
Overview of the diffractograms of the various materials identified by XRD. RBC-0 confirms its amorphous nature, whereas T-DCPD addition shows a clear peak pattern due to the dicalcium phosphate dihydrate (brushite, B). Incubation under acidic and biofilm conditions showed the conversion of brushite into monetite (M), the anhydrous salt.

**Figure 9 jfb-13-00232-f009:**
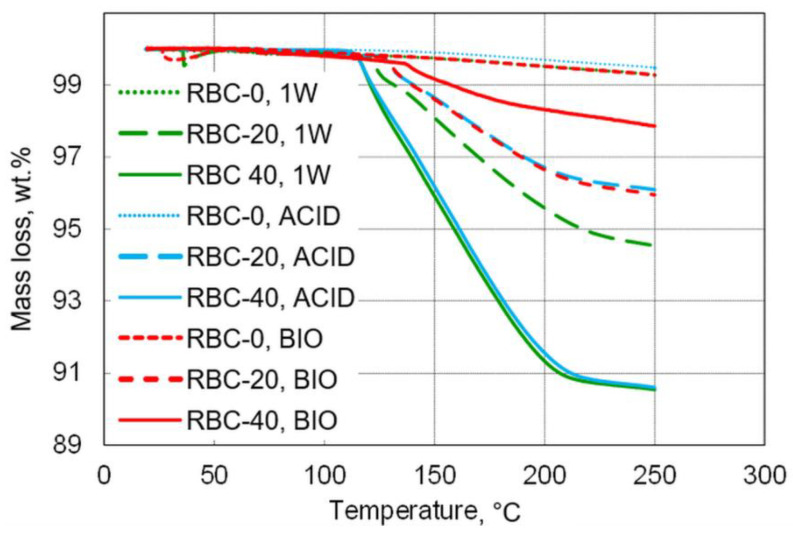
Overview of mass loss during thermogravimetric measurements up to 250 °C. T-DCPD incorporation led to an increase in mass loss, whereas, contrary to incubation under acidic conditions, biofilm incubation reduced mass loss.

**Figure 10 jfb-13-00232-f010:**
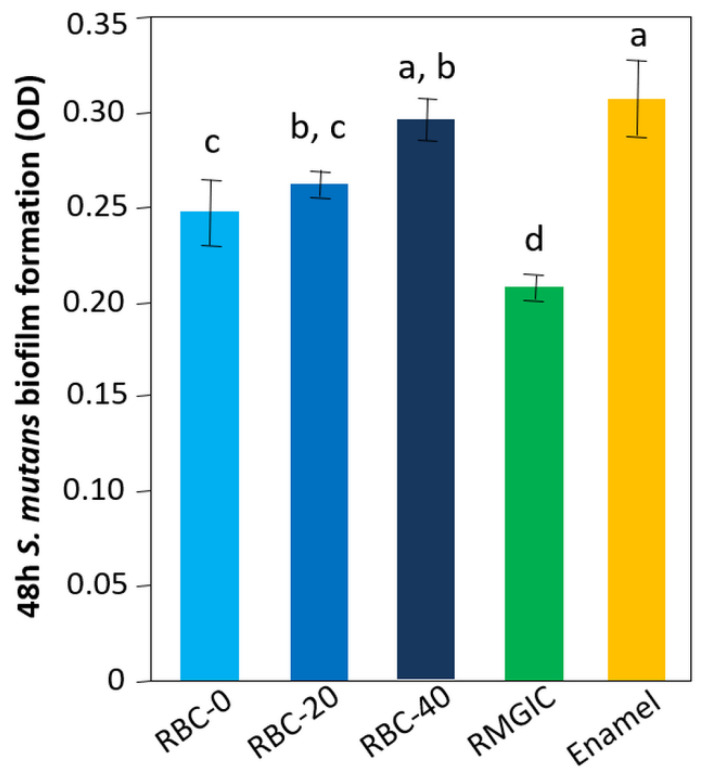
Viable biomass on the surface of the various test specimens after 48 h incubation. Means and standard deviations are indicated; different letters indicate significant differences between the various materials (*p* < 0.05).

**Figure 11 jfb-13-00232-f011:**
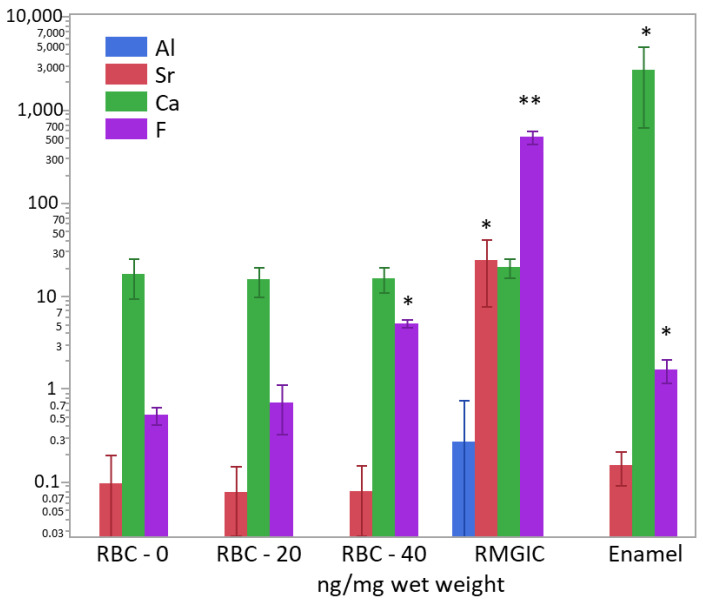
Content of Al^3+^, Sr^−^, Ca^2+^, and F^−^ in the biofilms (ng/mg wet weight) on the surface of the tested materials after 48 h incubation. Means and standard deviations are indicated. For sake of clarity, the highest level on the graph (10^4^ ng/mg) corresponds to 1 wt.%. Asterisks indicate significant differences between groups for each element (Tukey’s test, *p* < 0.05).

**Figure 12 jfb-13-00232-f012:**
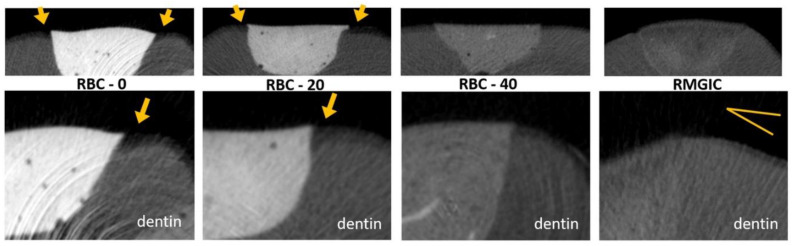
MicroCT coronal reconstructions of the interface between tested materials and dental tissues at the dentin level. Whole restorations and a close-up of one margin are displayed. Class II cavities were 4 mm wide and 2 mm deep. High demineralization depths were observed in RBC-0 and RBC-20 margins, two times deeper than in RBC-40 and RMGIC. Evidence of protection against demineralization in RBC-40 and RMGIC on margins < 1 mm from restoration were observed.

**Figure 13 jfb-13-00232-f013:**
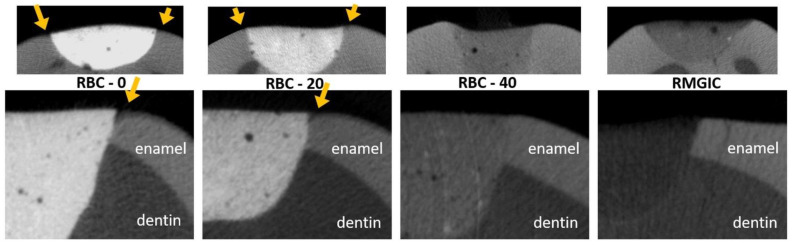
MicroCT coronal reconstructions of the interface between tested materials and dental tissues at the enamel level. Whole restorations and a close-up of one margin are displayed. Barium glass inert filler addition to experimental RBCs was responsible for the increased radiopacity and helped make the materials immediately identifiable. RBC-40, with the lowest barium content, had similar radiopacity to enamel, whereas the control RMGIC had similar radiopacity to dentin. RBC-0 and RBC-20 did not prevent enamel demineralization close to the restoration, which is a sign of initial secondary caries formation.

**Table 1 jfb-13-00232-t001:** Overview of the crystalline phases identified by X-ray diffraction on the surface of the various materials. DCPD (brushite), CaH[PO_4_]·2H_2_O; monetite, CaH[PO_4_].

Material, Treatment	Crystalline Phases
RBC-0, 1W	X-ray amorphous
RBC-20, 1W	DCPD (+)
RBC-40, 1W	DCPD (++)
RBC-0, ACID	X-ray amorphous
RBC-20, ACID	DCPD (+) monetite (+)
RBC-40, ACID	DCPD (++) monetite (++)
RBC-0, BIO	X-ray amorphous
RBC-20, BIO	DCPD (+) monetite (+)
RBC-40, BIO	DCPD (++) monetite (+)

**Table 2 jfb-13-00232-t002:** Content of DCPD and free water, as identified by TGA. Results expressed as wt.%.

Material, Treatment	Free Water Content (20–105 °C)	DCPD Content (105–230 °C)
RBC-0, 1W	0.1	0.0
RBC-20, 1W	0.5	22.1
RBC-40, 1W	0.1	40.2
RBC-0, ACID	0.0	0.0
RBC-20, ACID	0.2	15.4
RBC-40, ACID	0.0	34.3
RBC-0, BIO	0.1	0.0
RBC-20, BIO	0.2	5.5
RBC-40, BIO	0.2	14.2

**Table 3 jfb-13-00232-t003:** Supernatant pH changes induced by the acidic challenge. The materials were exposed to a pH = 4.2 solution obtained by filter-sterilizing the effluent from the continuous-flow bioreactor. In this way, specimens were exposed to the same solution with which they were challenged inside the bioreactor model (ACID and BIO specimens). Enamel values are shown as reference. Different superscript letters indicate significant differences between groups (Tukey’s HSD test, *p* < 0.05).

Material	pH (±1SD)
RBC-0	4.27 (0.01) ^d,e^
RBC-20	4.38 (0.01) ^d^
RBC-40	4.81 (0.03) ^c^
RMGIC	4.96 (0.05) ^b^
Enamel	5.48 (0.20) ^a^
Effluent	4.20 (0.00) ^e^

**Table 4 jfb-13-00232-t004:** Mean demineralization depth (µm ± 1SD) of the tooth specimens in dentine. Different superscript letters indicate significant differences between cavities restored with the tested materials.

Material	Margin	Margin + 1.0 mm
RBC-0	214.5 (44.9) ^a^	163.4 (29.8) ^a^
RBC-20	156.5 (42.8) ^a^	157.9 (58.1) ^a^
RBC-40	80.8 (33.6) ^b^	144.8 (34.6) ^a^
RMGIC	84.4 (47.6) ^b^	148.8 (41.6) ^a^

## Data Availability

The datasets generated during and/or analyzed during the current study are available from the corresponding author upon reasonable request.

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
