# Peer review of "TEGDMA-Functionalized Dicalcium Phosphate Dihydrate Resin-Based Composites Prevent Secondary Caries in an In Vitro Biofilm Model"

_jfb, 2022, doi:10.3390/jfb13040232_

Round 1
Reviewer 1 Report
Dear Authors
the content of the article raises no doubts about the selection of the methods used and the final conclusions. The study is original, and a wide selection of methods was used (the use of the biofilm flow model is particularly interesting - although it is not a novelty, and tomography for three-dimensional imaging of changes in the structure of fillings). As for the issues related to the structure of the text itself, it seems that by mistake the information about the bioethical commission was put in the wrong place (it is in chapter 2.2 and probably should be in 2.3).
With Best Regards
Author Response
We agree with the Reviewer and we thank him/her for the appreciation. According to the suggestion, the info about the commission was moved to the right place.
Reviewer 2 Report
Dear authors, I have really enjoyed reading this article, the subject is very important and well-described, as well as, the results are appropriately presented.
Comment 1. Please, in the methodology section of the secondary caries model, place the reference of said methodology.
Comment 2. Please, edit the caption of each micrograph in Figure 3, I can´t see the numbers for magnification and scale bar measurement.
Comment 3. The quality of Figure 5 makes it very complex to observe each chemical element. Please, improve the quality of the image. The image is composed of too many individual images. The images where each chemical element is colored (red, blue, green, yellow, etc) would be removed, so that the other images can be seen more clearly and larger.
Author Response
Dear authors, I have really enjoyed reading this article, the subject is very important and well-described, as well as, the results are appropriately presented.
Comment 1. Please, in the methodology section of the secondary caries model, place the reference of said methodology.
The following reference was added as per Reviewer’s suggestion.
Microbiological models for accelerated development of secondary caries in vitro Andrei C. Ionescu, Sebastian Hahnel, Paolo Delvecchio, Nicoleta Ilie, Marioara Moldovan, Vanessa Zambelli, Giacomo Bellani, Eugenio Brambilla. Journal of Dentistry Volume 127, December 2022, 104333 https://doi.org/10.1016/j.jdent.2022.104333
Comment 2. Please, edit the caption of each micrograph in Figure 3, I can´t see the numbers for magnification and scale bar measurement.
We agree with the Reviewer. The figure was a low-resolution version for the manuscript file. The original figure has all the needed resolution and details.
Comment 3. The quality of Figure 5 makes it very complex to observe each chemical element. Please, improve the quality of the image. The image is composed of too many individual images. The images where each chemical element is colored (red, blue, green, yellow, etc) would be removed, so that the other images can be seen more clearly and larger.
According to Reviewer’s suggestions, the figure was changed to have increased readability. Please note the high resolution figures read even better.
Reviewer 3 Report
Dear Authors,
you made a great work! However, some improvements are mandatory before acceptance.

Author Response
The paper is a original research on the TEGDMA-functionalized dicalcium phosphate dihydrate resin-based composites prevent secondary caries in an in-vitro bio- film model.
The Authors made a great work in terms of methodology and the paper sounds scientific and well written.
However, some improvements are mandatory before acceptance.
The abstract is well written, complete and summary in its various aspects. The keywords are complete and appropriate. Pease follow the abstract indications provided by MDPI.
In the introduction:
- “A previous study from our research group [22] compared biofilm formation on the surfaces of experimental RBCs featuring triethyleneglycol-dimethacrylate (TEGDMA)-functionalized or non-functionalized DCPD particles (T-DCPD) with the behavior of conventional RBCs.”
to enrich the introduction from this point of view, I suggest the authors also evaluate the secondary effects, having declared that these cements favor the deposition of minerals at the dentinal level, thanks also to the stem stimulation and the subsequent application of material as a response to a stimulus activating, as underlined by:
“Bhandi S, Alkahtani A, Reda R, Mashyakhy M, Boreak N, Maganur PC, Vishwanathaiah S, Mehta D, Vyas N, Patil V, Raj AT, Testarelli L, Patil S. Parathyroid Hormone Secretion and Receptor Expression Determine the Age-Related Degree of Osteogenic Differentiation in Dental Pulp Stem Cells. J Pers Med. 2021 Apr 27;11(5):349. doi: 10.3390/jpm11050349.”
“Bhandi S, Alkahtani A, Mashyakhy M, Abumelha AS, Albar NHM, Renugalakshmi A, Alkahtany MF, Robaian A, Almeslet AS, Patil VR, Varadarajan S, Balaji TM, Reda R, Testarelli L, Patil S. Effect of Ascorbic Acid on Differentiation, Secretome and Stemness of Stem Cells from Human Exfoliated Deciduous Tooth (SHEDs). J Pers Med. 2021 Jun 22;11(7):589. doi: 10.3390/jpm11070589.”
The Authors after careful consideration agree that adding such enrichment to the introduction does not help in providing a specific connection with the topic of the presented set of experiments.
Materials and methods are clear and well explained. Different aspects are analyzed with a dedicated statistical test. The authors did a great job in the explication of all the variables identified and included in the study.
Results are easy to understand and comprehensive. All the studied characteristics were reported in tables which are clear and concise.
Discussion: this section is complete and evaluates the outcome of different papers present in literature. The overall is comprehensive, concise and complete in its various aspects.
Conclusions are concise and clear.
Bibliography should be formatted respecting the journal’s requirements and no improper citations are evidenced.
The Reference section was thoroughly reviewed. Thank you.
Figures and labels are clear and easy to comprehend.
English is clear and easy to understand.
Reviewer 4 Report
I would like to thank the authors for their work and submission. In this work the authors have evaluated the demineralisation prevention efficacy of an experimental composite resin material containing Dicalcium Phosphate Dihydrate (DCPC) in comparison with Resin Modified Glass Ionomer (RMGI). The strength of the work is that it is well presented and several techniques including SEM, microCT, EDS and antimicrobial assays have been employed to clarify the research question. Yet, there are several points which need to be clarified to prevent making premature conclusion about the results.
As authors know, caries formation is a multifactorial process which involves various variables. In the case of secondary caries, factors such as the condition of the substrate (including degree of enamle and dentine mineralisation, smoothness of preparation edges etc), the restorative material factors (including the bond strength between the restorative material and the substrate, polymerisation shrinkage of the restorative material etc) and the overlying biofilm have significant effect on the prevention or formation of secondary caries. These factors should be considered and discussed. For example, one possible explanation for lower demineralisation in RBC40 compared to RBC 20 and RBC could be the lower amount of polymerisation shrinkage due to higher filler content and lower resin fraction. Polymerisation shrinkage has a direct effect on the formation of micro-gaps at the tooth-restoration interface which results in the development of secondary caries. The other point which needs to be clarified is that if etching and bonding have been used prior to filling the cavities. From the text, I implied that etching and bonding have not been used which in that case should be justified and explained (including the effect of using no bond and no etch on the formation of microgap and the secondary caries). In addition, the readers should be informed that high amount of filler content may adversely(?) affect the esthetics, workability, mechanical property and other features of composite resin and therefore need to be further investigated.
Overall, the work has merits for publications and would be of interest to the readers.
Author Response
Thank you for your appreciation.
We acknowledge the points that were raised by the Reviewer. Many of these factors were addressed, such as, for instance, smoothing the restorations’ and nearby tissues’ surfaces using a standardized procedure (Pag. 4, line 153). The degree of enamel and dentine mineralization is one of the factors associated with the variability of results when using natural substrates. This was better addressed in the methods section. As a pilot study was preliminarily performed to assess the effect of such variability on the outcome, the sample size was based on the available data. This specification was added to the text.
Regarding the effect of filler content on polymerization shrinkage, the materials’ composition was specified in the text and all tested experimental compositions had exactly the same filler-to-resin ratio. The differences in T-DCPD content were compensated for the filler using barium glass, as was confirmed by EDS analysis.
To focus on the effects of the experimental composite without interference from adhesive, no etching or adhesive procedures were used. The absence of micro-gaps was checked by Micro-CT analysis before cariogenic challenge (T=0). The analysis showed no evidence of microgaps in any of the tested restorations of the present study, in spite of not using any adhesive procedure/conditioning of tooth structures. Their use without an adhesive procedure allows comparison of the obtained results with literature data, see for instance reference # XX. Further experiments will make use of an adhesive procedure for better approaching clinical conditions. This topic was addressed in the discussion section.
Braga and Alania 2019; Melo 2013; Zhang, K., Zhang, N., Weir, M. D., Reynolds, M. A., Bai, Y., & Xu, H. H. (2017). Bioactive dental composites and bonding agents having remineralizing and antibacterial characteristics. Dental Clinics, 61(4), 669-687.
Regarding the mechanical properties and aesthetics of calcium phosphate-filled composite materials, the following sentence was added to the discussion section.
Filling an RBC material with a source of calcium and phosphate ions may represent a good choice in this sense, providing a material that may reduce secondary caries occur-rence. Nevertheless the material may still need improvements of its mechanical, physical, and esthetical properties [14,16]. It is especially difficult to manipulate the refractive indices of crystalline calcium phosphates, which hinders achieving translucency and good final aesthetics.
Reviewer 5 Report
October, 19, 2022
Dear authors
Thank you for an interesting report.
In this study, you examined the efficacy of TEGDMA-functionalized dicalcium phosphate dihydrate filler-based resin-based composites in preventing caries lesions around the restoration margins. Resin-based composites have been important restorative materials in dentistry for many years. Following previous improvements in this series of materials, the addition of more aggressive secondary caries protection is expected to further improve patient satisfaction.
I agree to many parts of your claims and guessed that the subject of this paper will be of interest to the readership of Journal Functional Biomaterials. However, I think that minor revisions are required as follows:
Throughout this paper
1. As for “TEGDMA-functionalized dicalcium phosphate dihydrate filler”, I think that it is necessary for you to roughly explain the structure of "TEGDMA-functionalized dicalcium phosphate dihydrate filler" and how it exists in the resin composite in an easy-to-understand manner using schematic diagrams. Although the preparation method is described, readers do not have an image of the chemical structure of TEGDMA-functionalized dicalcium phosphate.
Materials and Methods
1. Many readers find it difficult to understand because there are many explanations in sentences. I think that you can make the presentation easier by incorporating flow charts, other figures, tables, etc. as appropriate to make it easier for the reader to read, that is, easier to understand.
2. I think that it would be better if you included images of test pieces before and after biofilm formation.
Results
1. In Figs. 3 to 5, you partially describe which surface of the test piece each SEM image is, but it is unclear and difficult to see from the reader's side. I think that you should display the code of the test piece and the magnification more clearly.
2. In Figs 2, 10, and 11, I think that it would be easier for the readers to understand if you insert scales in the micro-CT images.

Author Response
Dear authors
Thank you for an interesting report.
In this study, you examined the efficacy of TEGDMA-functionalized dicalcium phosphate dihydrate filler-based resin-based composites in preventing caries lesions around the restoration margins. Resin-based composites have been important restorative materials in dentistry for many years. Following previous improvements in this series of materials, the addition of more aggressive secondary caries protection is expected to further improve patient satisfaction.
I agree to many parts of your claims and guessed that the subject of this paper will be of interest to the readership of Journal Functional Biomaterials. However, I think that minor revisions are required as follows:
Thank you very much. The manuscript was revised and suggestions addressed, as follows.
Throughout this paper
- As for “TEGDMA-functionalized dicalcium phosphate dihydrate filler”, I think that it is necessary for you to roughly explain the structure of "TEGDMA-functionalized dicalcium phosphate dihydrate filler" and how it exists in the resin composite in an easy-to-understand manner using schematic diagrams. Although the preparation method is described, readers do not have an image of the chemical structure of TEGDMA-functionalized dicalcium phosphate.
We agree with the Reviewer. The presence of a functionalizing TEGDMA layer grafted on the particles’ surface was described in the diagram that was inserted in the manuscript with the following explanation.
Diagram illustrating the DCPD-TEGDMA interaction. The DCPD structure is represented by the calcium ions and the hydrogen phosphate tetrahedra. At the DCPD-TEGDMA interface, two triethylene glycol groups from TEGDMA are represented, with ion-dipole bonds established between one of their oxygen atoms and the calcium ion.
Materials and Methods
- Many readers find it difficult to understand because there are many explanations in sentences. I think that you can make the presentation easier by incorporating flow charts, other figures, tables, etc. as appropriate to make it easier for the reader to read, that is, easier to understand.
Considering Reviewer’s suggestion, a flow-chart describing the used methodology was added to the methods section.
- I think that it would be better if you included images of test pieces before and after biofilm formation.
We examined the test pieces after biofilm formation, however these images did not add any information valuable to the reader. Micro-CT images provided a much better insight of the effect of biofilm formation on the test specimens.
Results
- In Figs. 3 to 5, you partially describe which surface of the test piece each SEM image is, but it is unclear and difficult to see from the reader's side. I think that you should display the code of the test piece and the magnification more clearly.
Figure 5 was modified for improved readability as also suggested by other Reviewers. Please note that the manuscript contains low-resolution of the figures that in the final manuscript will be in high-resolution and more clear.
- In Figs 2, 10, and 11, I think that it would be easier for the readers to understand if you insert scales in the micro-CT images.
We agree with the Reviewer. The readers cannot understand the cavity dimensions in the Micro – CT images. The following sentence was added to the methods section, and the cavity dimensions were specified in the figures captions to provide readers with this piece of information.
A total of four standardized class II cavities (4 mm width, 2 mm depth, 6 mm height), one for each side, were prepared on each tooth specimen using 2 mm-wide cylindrical diamond burs. The cervical margins of the cavities were located in the dentin.